# Effect of Switching to Once-Weekly Semaglutide on Non-Alcoholic Fatty Liver Disease: The SWITCH-SEMA 1 Subanalysis

**DOI:** 10.3390/pharmaceutics15082163

**Published:** 2023-08-20

**Authors:** Hiroshi Nomoto, Yuka Takahashi, Yoshinari Takano, Hiroki Yokoyama, Kazuhisa Tsuchida, So Nagai, Aika Miya, Hiraku Kameda, Kyu Yong Cho, Akinobu Nakamura, Tatsuya Atsumi

**Affiliations:** 1Department of Rheumatology, Endocrinology and Nephrology, Faculty of Medicine and Graduate School of Medicine, Hokkaido University, Sapporo 060-8638, Hokkaido, Japan; 2Department of Diabetes and Endocrinology, Tonan Hospital, Sapporo 060-0004, Hokkaido, Japan; 3Jiyugaoka Medical Clinic, Obihiro 080-0016, Hokkaido, Japan; 4Manda Memorial Hospital, Sapporo 060-0062, Hokkaido, Japan; 5Division of Diabetes and Endocrinology, Department of Medicine, Sapporo Medical Centre, NTT East Corporation, Sapporo 060-0062, Hokkaido, Japan

**Keywords:** fatty liver, glucagon-like receptor agonists, once-weekly semaglutide, type 2 diabetes

## Abstract

Non-alcoholic fatty liver disease (NAFLD) is an important common comorbidity in individuals with type 2 diabetes (T2DM). Although some glucagon-like peptide-1 receptor agonists (GLP-1RAs) have beneficial effects on NAFLD, the efficacy of once-weekly semaglutide has not been established. This was a subanalysis of the SWITCH-SEMA 1 study, a multicenter, prospective, randomized, parallel-group trial comparing switching from liraglutide or dulaglutide to once-weekly semaglutide in subjects with T2DM (SWITCH) versus continuing current GLP-1RAs (Continue) for 24 weeks. This subanalysis consisted of participants who were suspected to have NAFLD [fatty liver index (FLI) ≥ 30]. In total, 58 participants met the criteria of this subanalysis. There were no statistical differences in baseline characteristics between the SWITCH (*n* = 31) and Continue groups (*n* = 27). FLI significantly improved during treatment in the SWITCH group (68.6 to 62.7) but not in the Continue group (71.1 to 72.3) (*p* < 0.01). The improvement of FLI in the SWITCH group was greater in switching from dulaglutide to semaglutide and significantly correlated with older age (*p* = 0.016) and lower baseline FLI (*p* < 0.01). The multiple linear regression analysis revealed that the switch from dulaglutide was associated with an improvement in FLI (*p* = 0.041). Switching from conventional GLP-1RAs to once-weekly semaglutide might be beneficial for individuals with NAFLD complicated with T2DM.

## 1. Introduction

The ultimate goal of treating type 2 diabetes (T2DM) is to maintain quality of life and life expectancy at a comparable level as that in subjects without diabetes [1]. To achieve this goal, both managing metabolic dysfunction and preventing diabetic complications caused by hyperglycemia are required [1,2]. Although numerous anti-hyperglycemic agents are available, regimens with beneficial effects on impaired organs should be considered according to each comorbidity [3].

Non-alcoholic fatty liver disease (NAFLD) is an important common comorbidity in individuals with T2DM. The all-cause mortality and incidence of liver-related complications increase with worsening liver fibrosis in subjects with NAFLD, and subjects with progressive fibrosis have a higher incidence of T2DM [4]. In addition, among several metabolic abnormalities, T2DM was identified as a significant risk factor for the development of non-alcoholic steatohepatitis (NASH) and the progression of liver fibrosis in women with NASH [5]. Managing glycemic control and body weight is considered critical for improving such histological changes in the liver [6]; however, validated pharmacological approaches are limited for NAFLD/NASH [7]. For NAFLD complicated with T2DM, the use of vitamin E, pioglitazone, glucagon-like peptide-1 receptor agonists (GLP-1RAs), and sodium–glucose cotransporter-2 (SGLT2) inhibitors are recommended as a drug therapy [8]. Among these, GLP-1RAs and SGLT2 inhibitors are frequently used in daily clinical practice, but it is important to establish a treatment strategy for NAFLD that is poorly managed even under the use of these agents.

Although some GLP-1RAs have beneficial effects on NAFLD, the efficacy of once-weekly semaglutide, especially after a switch from other GLP-1RAs, has not been established. Therefore, we evaluated the effects of semaglutide on NAFLD based on our previous study assessing the efficacy of semaglutide compared with that of other GLP-1RAs in subjects with T2DM [9]. Because the original study did not perform imaging or histopathology of the liver, we assessed indices reflecting NAFLD [fatty liver index (FLI), hepatic steatosis index (HSI), and Zhejiang University (ZJU) index] [10,11,12]. Among them, we focused especially on the changes in FLI between the different treatment strategies because of its usefulness in subjects with T2DM and Japanese subjects [13,14].

## 2. Materials and Methods

### 2.1. Study Design and Participants

This was a secondary analysis of our previous multicenter prospective open-labeled, randomized, parallel-group comparison study comparing the efficacy of switching from liraglutide or dulaglutide to once-weekly semaglutide on glycemic control in Japanese adults with T2DM [9]. In the original trial, non-lean adults [20–90 years and body mass index (BMI) ≥ 22 kg/m^2^] with T2DM and glycated hemoglobin (HbA1c) levels of 6.0–10.0% on treatment with liraglutide (0.9–1.8 mg/day) or dulaglutide (0.75 mg/week) for more than 12 weeks were recruited from nine sites in Hokkaido. Written informed consent was obtained from all participants. The exclusion criteria were as follows: (1) treated with GLP-1RAs other than liraglutide and dulaglutide; (2) allergy to semaglutide; (3) unstable diabetic retinopathy; (4) severe hepatopathy or nephropathy; (5) severe ketosis, diabetic coma, or precoma; (6) severe infection, trauma, and/or recent or planned surgery; (7) definite or suspected pregnancy; and (8) incompatibility with the trial for other reasons as determined by the physician as described previously [9]. There were no restrictions concerning the concomitant use of medications that could affect the pathophysiology of NAFLD at baseline. Changes in treatment for comorbidities were prohibited in principle during the study period. The participants were randomly assigned to continue their current GLP-1RA therapy or switch to once-weekly semaglutide (0.25–1.0 mg/week). The starting dose of semaglutide was 0.25 mg/week, and after at least 4 weeks, the dose was increased to 0.5–1.0 mg/week, as described previously [9]. The final dose of semaglutide was decided by the physicians in charge based on the condition of each participant. In the semaglutide group, the final dose for semaglutide was determined at the discretion of each attending physician basically according to the accompanying text. Diet and exercise regimens were continued appropriately at each site, and principally, medications for comorbidities were not changed during the study period. Glycemic control indices, fasting serum/urine biomarkers, physical assessment markers, and treatment satisfaction, as assessed via a diabetes treatment satisfaction questionnaire, were evaluated at baseline and at the end of this study (24 weeks). As a marker of the presence or extent of fatty liver, the FLI, consisting of BMI, triglyceride (TG), γ-glutamyl transpeptidase (γ-GTP), and waist circumference (WC), was calculated using the following formula: FLI = {exp [0.953 × log (TG) + 0.139 × BMI + 0.718 × log (γ-GTP) + 0.053 × WC − 15.745]/1 + exp [0.953 × log (TG) + 0.139 × BMI + 0.718 × log (γ-GTP) + 0.053 × WC − 15.745]} × 100 [10]. Similarly, HSI and ZJU index were investigated using following equations for confirmation: HIS = 8 × [alanine aminotransferase (ALT)/aspartate aminotransferase (AST) ratio] + BMI (+2, if female; +2, if diabetes mellitus) [11]; ZJU index = BMI + fasting plasma glucose (mmol/L) + TG (mmol/L) + 3 × (ALT/AST ratio) (+2, if female) [12]. In addition, hepatic fibrosis was estimated the Fibrosis-4 (FIB-4) index, which was derived as follows: FIB-4 index = age × [AST/(platelet count × ALT)^1/2^] [15]. Because we could not obtain the data regarding the precise alcohol consumption of the participants from the original study, habitual drinkers were excluded to minimize the effect of alcohol on fatty liver in this analysis. In addition, patients with FLI < 30 were also excluded considering that this cutoff can be used to eliminate the possibility of hepatic steatosis. The primary outcome of this subanalysis was the impact of continuing GLP-1RAs and switching to semaglutide on indices reflecting NAFLD. Second, factors associated with changes in FLI were explored.

The SWITCH-SEMA-1 study was registered with the Japan Registry of Clinical Trials (jRCTs1011200008). The study protocol was approved by the Hokkaido University Certified Review Board (CRB no. 1180001), and this study was conducted in accordance with the principles of the Declaration of Helsinki and its amendments. All participants provided written informed consent before participation.

### 2.2. Statistical Analysis

Normally distributed data were expressed as the mean ± SD. Non-normally distributed variables were expressed as the median (quartiles), and categorical variables were expressed as *n* (%). Differences between the two groups were compared using an unpaired *t*-test for normally distributed continuous variables and the Mann–Whitney U-test for other variables. The chi-squared test or Fisher’s exact test was applied for categorical variables. The results were compared between the groups using a paired *t*-test for normally distributed data or Wilcoxon’s signed-rank test for other variables. The Shapiro–Wilk test was used to determine the appropriate statistical test for the continuous variables. Because the effects of GLP-1RAs on FLI and the FIB-4 index can be affected by each baseline variable and patients’ background data, we also conducted analysis of covariance to adjust for these confounders. Correlations were evaluated using Spearman’s rank correlation analysis. Multivariate analyses were performed using multiple linear regression to identify factors independently associated with the outcomes. Data were analyzed using GraphPad Prism 8.4.2 (GraphPad Software, Inc., San Diego, CA, USA) or JMP Pro 16.0.0 (SAS Inc., Cary, NC, USA). The post hoc power calculation was performed using GPower^®^ version 3.1.9.2. *p* < 0.05 indicated statistical significance.

## 3. Results

Of 100 participants from the original cohort, 42 were excluded from this subanalysis for the following reasons: habitual drinking (*n* = 16), FLI < 30 (*n* = 20), a medical history of chronic hepatitis B (*n* = 3), the addition of pemafibrate (*n* = 1), and lack of relevant data (*n* = 2). Therefore, 27 subjects who continued current GLP-1RA therapy (Continue) and 31 subjects who changed their treatment regimens to semaglutide (SWITCH) were analyzed (Figure 1). As a result, mean age, BMI, and HbA1c were 58.9 ± 12.4 years old, 31.0 ± 4.9 kg/m^2^, and 7.9 ± 0.8%, respectively. There were no significant differences in baseline characteristics between the groups, including background treatments for T2DM and FLI (Table 1). At baseline, liraglutide was administered at a dose of 0.9 mg/day or higher in all participants, while dulaglutide was used at a dose of 0.75 mg/week, the only dose approved in Japan. Most study participants were treated with metformin and/or SGLT2 inhibitors, with no cases using dipeptidyl peptidase-4 inhibitors due to the study design.

After switching to semaglutide, glycemic control indices and metabolic parameters, including BMI and liver enzymes, improved as observed in the original cohort [fasting plasma glucose: −16.0 mg/dL (*p* = 0.014); HbA1c: −0.8% (*p* < 0.001); BMI: −0.87 kg/m^2^ (*p* < 0.001); ALT: −5 IU/L (*p* = 0.018); γ-GTP: −3 IU/L (*p* = 0.08)] [9]; on the other hand, no such changes were confirmed in the Continue group (Table 2). FLI, the main outcome of this subanalysis, significantly improved in the SWITCH group (from 68.6 ± 24.5 to 62.7 ± 28.0, *p* = 0.002) but not in the Continue group (from 71.1 ± 18.5 to 72.3 ± 19.5, *p* = 0.490), resulting in a significant difference between the groups (*p* = 0.005, Figure 2). This result was also borne out by two other different indices for fatty liver, the HSI and the ZJU index (HSI in SWITCH group: from 43.5 ± 7.4 to 41.7 ± 7.3, *p* < 0.001, ZJU index in SWITCH group: 44.8 ± 6.3 to 42.6 ± 6.0, *p* < 0.001) (Appendix A). The post hoc power calculation illustrated that the overall detection power value for changes in FLI between continuing current GLP-1RAs and switching to semaglutide under the 5% α error was 0.825. Then, we conducted an ANCOVA adjusted for baseline each index, HbA1c, and BMI and confirmed that the results were robust (Appendix A). Conversely, switching to semaglutide did not ameliorate liver fibrosis as assessed by the FIB-4 index (Appendix A). Interestingly, positive correlations between changes in FLI and those in indices for glycemic control and liver deviate enzymes were observed in the Continue group [correlation with changes in fasting plasma glucose (ρ = 0.479, *p* = 0.012), HbA1c (ρ = 0.418, *p* = 0.030), AST (ρ = 0.351, *p* = 0.073), and ALT (ρ = 0.353, *p* = 0.071)], indicating a close relationship of metabolic parameters with FLI; however, such correlations were not detected in the SWITCH group (Appendix A).

To identify patients with NAFLD who can benefit from a switch to semaglutide, we examined the patients’ background factors associated with improvements in FLI. Correlation analysis revealed that older age and lower baseline FLI was significantly correlated with improvement in FLI in the SWITCH group (*p* = 0.016 and 0.005, respectively, Table 3). Importantly, baseline glycemic control and liver and/or kidney function were not correlated with the effects of semaglutide on FLI. In addition, participants who changed their treatment regimens from dulaglutide to semaglutide showed larger improvements in FLI than those who changed regiments from liraglutide (Figure 3, Appendix A). Multivariate analysis using multilinear regression revealed that only a switch from dulaglutide was positively correlated with improvement in FLI, and age and baseline FLI were no longer significantly correlated (Appendix A). Considering the correlation between baseline GLP-1RA regimens and changes in FLI in the SWITCH group, we next compared the baseline characteristics between two switching strategies: from liraglutide to semaglutide and from dulaglutide to semaglutide. As presented in Appendix A, background patient characteristics including baseline FLI did not differ between these strategies. Notably, both switch strategies resulted in significant reductions in HbA1c and BMI [HbA1c: −0.8% in switching from liraglutide (*p* < 0.001) vs. −0.9% in switching from dulaglutide (*p* < 0.001); BMI: −0.8 kg/m^2^ in switching from liraglutide (*p* = 0.019) vs. −1.0 kg/m^2^ in switching from dulaglutide (*p* < 0.001)], and there were no significant differences regarding the extent of the reduction between the subgroups, indicative of independent mechanisms of the switch from dulaglutide to semaglutide on FLI (Appendix A, Appendix A). The repeated measures of MANCOVA also showed the same tendency (Appendix A).

## 4. Discussion

In this subanalysis of a previous randomized controlled prospective trial, we focused on the efficacy of once-weekly semaglutide compared with that of other GLP-1RAs in patients with NAFLD and T2DM using an index reflecting the extent of fatty liver and liver fibrosis. All subjects were treated with liraglutide or dulaglutide for at least 12 weeks prior to inclusion, and approximately 90% of participants received concomitant SGLT2 inhibitors in this study. Our results illustrated that switching from current GLP-1RAs to once-weekly improved fatty liver indices, including FLI, HIS, and ZJU index. To the best of our knowledge, this is the first report directly comparing the effects of different GLP-1RAs on NAFLD with T2DM.

Avoiding poor glycemic control using appropriate anti-diabetic agents with proven effects on NAFLD and reducing excessive body weight are important components for preventing the progression to NASH and/or improving the pathophysiology of NAFLD [16,17]. In our study, all participants were pretreated with GLP-1RAs, which have clinical evidence of benefits against NAFLD. A synthetic analysis of phase 3 trials revealed that dulaglutide improved the ALT, AST, and γ-GTP levels compared with placebo in a pattern consistent with liver fat reduction [18]. On the other hand, a meta-analysis assessing the effects of 0.9–1.8 mg/day liraglutide on T2DM with NAFLD revealed favorable effects on both glycemic controls and metabolic abnormalities, including ALT [19]. Notably, these two injectable agents improved both serum biomarkers and liver fat content [20,21]. Even with such potent effects of GLP-1RAs on NAFLD, our study highlighted that managing metabolic abnormalities is important under GLP-1RA treatment (Appendix A).

Switching from other GLP-1RAs to semaglutide further improved glycemic control and several metabolic parameters [9], all of which can affect the pathophysiology and progression of NAFLD [22,23]. Indeed, FLI, HSI, and the ZJU index, which are used as surrogate measures of liver steatosis, were significantly improved with these switching strategies. A previous report described the better performance of FLI in diagnosing NAFLD than HSI, and its usefulness was demonstrated in the Japanese population. [14]. Although the ZJU index had a larger area under the ROC curve for detecting NAFLD than FLI and HSI [12], its usefulness in subjects with T2DM has been limited [24]. Therefore, we mainly focused on FLI in our analysis. Although the switching strategy improved FLI, the improvement in FLI was not correlated with changes in other metabolic parameters in the SWITCH group despite the remarkable effects of semaglutide on these indices. In addition, multivariate analysis demonstrated that switching from dulaglutide was the only independent factor related to improvements in FLI. Weekly semaglutide has been revealed to exert potent effects on glycemic control and body weight compared with other GLP-1RAs, including liraglutide and dulaglutide [9,25,26]. The proposed mechanisms of the effects of GLP-1RAs on NAFLD are as follows: (1) body weight loss, (2) improved glycemic control, (3) reduced insulin resistance, (4) increased lipolysis and fatty acid oxidation, and (5) the alleviation of ER stress and improved anti-inflammatory responses [27]. In this study, it was difficult to clarify whether the effects of semaglutide on NAFLD observed in our analysis were independent of the improved glycemic control and/or body weight loss because switching to semaglutide significantly improved both metabolic abnormalities in this study. However, it is worth noting that the dose of dulaglutide was fixed to 0.75 mg/week, which is the only dose approved in Japan, which possibly affected the results. As verified in a previous study, pharmacokinetic and pharmacodynamic differences exist among liraglutide, dulaglutide, and semaglutide [28]. Switching from liraglutide or dulaglutide to semaglutide resulted in greater serum GLP-1 concentrations, and when comparing liraglutide 1.2 or 1.8 mg/day with dulaglutide 0.75 mg/week, a higher GLP-1 concentration was confirmed for liraglutide [28]. Considering that the magnitude of the improvements in HbA1c and BMI was similar under both switching strategies (liraglutide to semaglutide or dulaglutide to semaglutide) in this subanalysis (Appendix A), potential differences in the underlying molecular mechanisms of liraglutide and dulaglutide, especially regarding intrahepatic metabolism caused by differences in the GLP-1 concentration, could exist.

Switching to semaglutide did not affect the results for the FIB-4 index, which reflects liver fibrosis. Recently, the efficacy of semaglutide versus placebo in GLP-1RA–naïve subjects with NASH with or without T2DM was evaluated, including a prospective double-blind phase 2 trial study design. Treatment with semaglutide resulted in a significantly higher resolution rate of NASH and improvement in the nonalcoholic fatty liver activity score, although liver fibrosis was not altered after semaglutide administration despite potent body weight reduction [29]. In contrast, our subanalysis focused on subjects with T2D who were treated with other GLP-1RAs, and our observation period was shorter than that of the phase 2 trial. In addition, liver fibrosis was lesser extent (Appendix A). Although switching to semaglutide improved FLI, our study duration was insufficient for evaluating changes in liver fibrosis considering that such effects were not confirmed even after 72 weeks of treatment [30].

The limitations of the original trial have been described previously [9], including a relatively short observation period, the inclusion of only Japanese participants, and a fixed dulaglutide dose of 0.75 mg/week. The present subanalysis had additional limitations mainly attributable to the study design of secondary analysis. The sample size might not have been sufficient for subgroup analysis because it was calculated to reveal differences in HbA1c levels between the groups, and we selected the subjects suspected of having NAFLD from the original cohort. However, there was no difference in patient background between the study groups, and the main findings of switching to semaglutide such as improved glycemic control, body weight, and metabolic parameters were similar to those in the original study. In addition, the baseline doses of dulaglutide and liraglutide might affect the efficacy of switching to semaglutide. The only approved dose of dulaglutide in Japan is 0.75 mg/week, and although the LEAN trial observed preferable effects on NASH for 1.8 mg/day liraglutide [30], most participants in our trial received the lower dose of liraglutide. Although we could not obtain data reflecting the pharmacokinetics of GLP-RAs in our study, differences in the serum GLP-1 concentration achieved via the GLP-1RAs used in this study could have affected the results. Finally, our evaluation focused on the indices of NAFLD, and pathological examination was not available. FLI is calculated using BMI, WC, TG, and γ-GTP, and it was difficult to evaluate the relationship between FLI and these individual factors. A further large-scale international randomized controlled trial including pathological assessments in patients with NAFLD is required in the future.

In summary, our study focused on the clinical approaches for subjects with NAFLD complicated with T2DM who received liraglutide or dulaglutide. Switching from these GLP-1RAs to semaglutide might have preferable effects on liver steatosis independent of glycemic control. Such switching could also improve metabolic abnormalities such as glycemic control and body weight management; however, its effects on liver fibrosis were limited. Although further studies are warranted, the enhancement of effects on GLP-1 by switching to semaglutide could represent a beneficial treatment option for subjects with various metabolic abnormalities, including NAFLD.

## Figures and Tables

**Figure 1 pharmaceutics-15-02163-f001:**
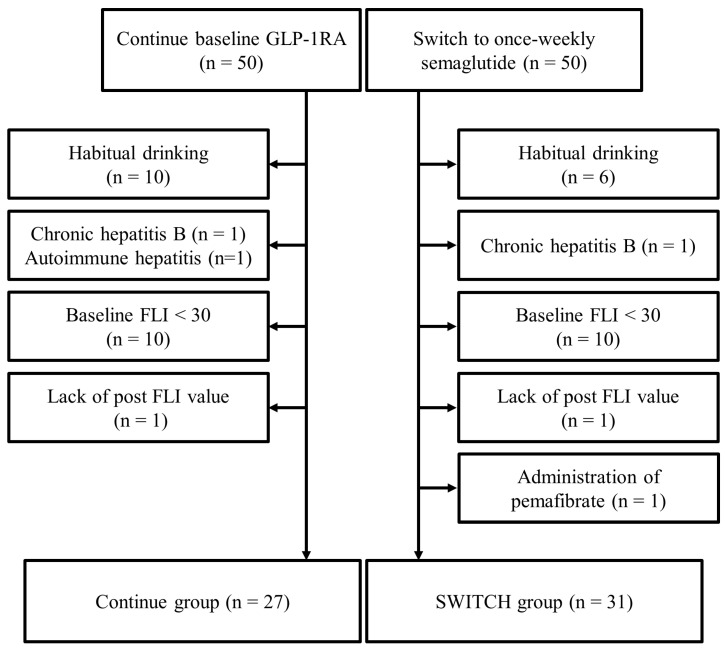
Flow diagram for the subanalysis. From the original cohort, subjects, diagnosed with liver disease other than non-alcoholic fatty liver disease, were habitual drinkers, had low baseline fatty liver index, and received drugs for comorbidities during the study period were excluded. FLI, fatty liver index; GLP-1RA, glucagon-like peptide-1 receptor agonist.

**Figure 2 pharmaceutics-15-02163-f002:**
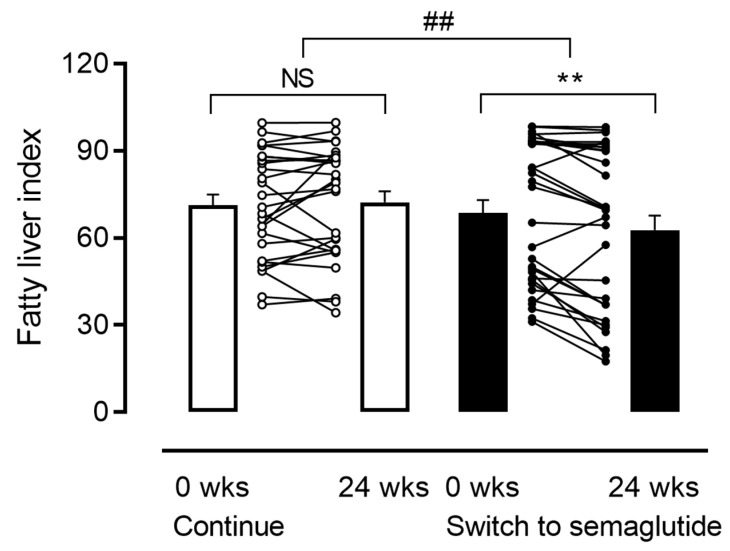
Changes in the fatty liver index from baseline. Changes in the fatty liver index during the 24-week study period (Continue group vs. SWITCH group). Bars represent the mean ± SE. ** *p* < 0.01 between 0 and 24 weeks, paired *t*-test. ## *p* < 0.01 between the Continue and SWITCH groups, unpaired *t*-test. 0 wks, baseline of this study; 24 wks, end of this study. NS, not significant; wks, weeks.

**Figure 3 pharmaceutics-15-02163-f003:**
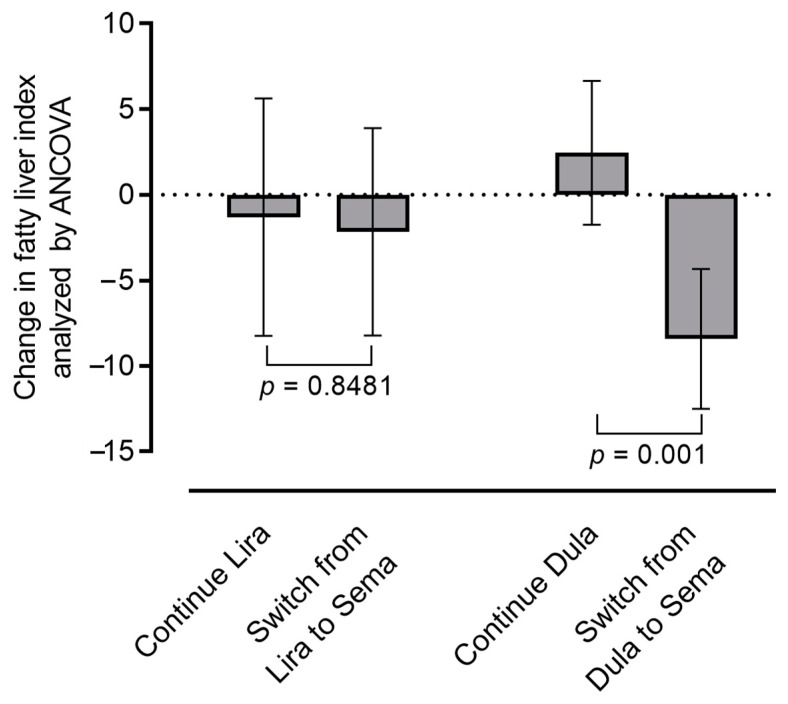
Comparison of the changes in the fatty liver index according to the baseline treatment regimens. Changes in the fatty liver index during the 24-week treatment period according to the baseline treatment regimens (switch from liraglutide or dulaglutide). Continue liraglutide (*n* = 10), switch from liraglutide to semaglutide (*n* = 13), continue dulaglutide (*n* = 17), switch from dulaglutide to semaglutide (*n* = 18). Data were adjusted for analysis of covariance (covariates: baseline fatty liver index, age, and body mass index). Bars represent the adjusted mean (95% confidence interval). Dula, dulaglutide; Lira, liraglutide; Sema, semaglutide.

**Table 1 pharmaceutics-15-02163-t001:** Baseline clinical characteristics.

Variables	Continue (*n* = 27)	SWITCH (*n* = 31)	*p*-Value
Age	56.3 ± 11.6	61.2 ± 12.8	0.131
Female sex, *n* (%)	15 (55.6)	12 (38.7)	0.292
Duration for diabetes, *n* (%)			0.435
<5 years	2 (7.4)	1 (3.2)	
5–15 years	11 (40.7)	9 (29.0)	
>15 years	14 (51.9)	21 (69.7)	
Current smoker, *n* (%)	9 (33.3)	8 (25.8)	0.574
Baseline GLP-1RA, *n* (%)			0.791
Liraglutide	10 (37.0)	13 (41.9)	
0.9 mg/day	3	6	
1.2 mg/day	5	3	
1.5 mg/day	1	2	
1.8 mg/day	1	2	
Dulaglutide	17 (63.0)	18 (58.1)	
0.75 mg/week	17	18	
Antihyperglycemic drugs, *n* (%)			
Metformin	25 (92.6)	28 (90.3)	1.000
SGLT2 inhibitors	24 (88.9)	27 (87.1)	1.000
Sulfonylureas	6 (22.2)	7 (22.6)	1.000
Glinides	5 (18.5)	2 (6.5)	0.233
Alpha-glucosidase inhibitors	5 (18.5)	2 (6.5)	0.233
Thiazolidinediones	3 (11.1)	7 (21.6)	0.311
Insulin treatment	11 (40.7)	13 (41.9)	1.000
Proportion of fatty liver index, *n* (%)			0.283
30 ≤ Fatty liver index < 60	8 (29.6)	14 (45.2)	
60 ≤ Fatty liver index	19 (70.4)	17 (54.8)	
Comorbidities, *n* (%)			
Hypertension	17 (63.0)	23 (74.2)	0.579
Dyslipidemia	23 (85.2)	26 (83.9)	1.000
Hyperuremia	5 (18.5)	6 (19.4)	1.000
Diabetic retinopathy, *n* (%)	7 (25.9)	9 (29.0)	1.000
Diabetic nephropaty, *n* (%)			
Microalbuminuria	5 (18.5)	7 (22.6)	0.756
Macroalbuminuria	4 (14.8)	5 (16.1)	1.000
Diabetic neuropathy, *n* (%)	5 (18.5)	9 (29.0)	0.378

Data are presented as the mean ± SD or *n* (%). *p*-value: The significance of differences between Continue group vs. SWITCH group, unpaired *t*-test, Fisher’s exact test, or chi-squared test. GLP-1RA, glucagon-like peptide-1 receptor agonist; SGLT2, sodium–glucose cotransporter-2.

**Table 2 pharmaceutics-15-02163-t002:** Changes in clinical parameters from baseline.

Variables	Baseline (Continue; *n* = 27)	Mean Change at 24 wks	Baseline (Switch to Semaglutide; *n* = 31)	Mean Change at 24 wks	*p*-Value
Body mass index (kg/m^2^)	30.8 ± 4.2	0.06 (−0.20 to 0.32)	31.1 ± 5.4	−0.85 (−1.20 to −0.59) ***	<0.001
Waist (cm)	103.4 ± 11.4	0.8 (−1.0 to 2.6)	105.4 ± 12.9	−1.8 (−4.1 to 0.4)	0.067
Systolic BP (mmHg)	128.4 ± 13.1	4.6 (−2.4 to 11.6)	129.6 ± 13.1	1.0 (−3.2 to 5.2)	0.356
Diastolic BP (mmHg)	80.6 ± 11.9	3.3 (0.0 to 6.5) *	79.0 ± 10.1	−3.1 (−5.6 to −0.7) *	0.002
FPG (mg/dL)	147.8 ± 40.3	−5.4 (−16.4 to 5.5)	145.8 ± 35.2	−16.0 (−28.6 to −3.5) *	0.205
HbA1c (%)	7.8 ± 0.9	0.1 (−0.1 to 0.3)	8.0 ± 0.7	−0.8 (−1.1 to −0.6) ***	<0.001
AST (IU/L)	26 (19–42)	−1 (−4 to 3)	21 (19–29)	−1 (−5 to 1)	0.707
ALT (IU/L)	41 (21–55)	−2 (−9 to 8)	27 (20–38)	−5 (−8 to 0) *	0.275
γ-GTP (IU/L)	32 (22–50)	1 (−3 to 3)	26 (17–48)	−3 (−5 to 0) **	0.056
eGFR (mL/min/1.73 m^2^)	70.9 ± 22.5	−1.4 (−3.4 to 0.6)	71.8 ± 22.6	−0.2 (−2.6 to 2.2)	0.450
Total cholesterol (mg/dL)	170.9 ± 31.9	−2.6 (−9.3 to 4.2)	174.0 ± 31.1	−11.7 (−20.4 to −3.0) **	0.101
Triglycerides (mg/dL)	137 (110–194)	0 (−24 to 20)	111 (80–162)	−10 (−24 to 20)	0.158
HDL-cholesterol (mg/dL)	49.2 ± 14.5	−1.4 (−4.8 to 0.7)	54.7 ± 13.3	−2.1 (−4.8 to 0.7)	0.734
TDI in insulin-treated patients ^a^	30 (15–78)	0 (0 to 2)	28 (24–37)	0 (−4 to 0)	0.123
Total DTSQ score ^b^	27.2 ± 4.8	−1.6 (−3.9 to 0.7)	25.9 ± 6.5	4.9 (2.1 to 7.7) **	<0.001

Data are presented as the mean ± SD or median (25th–75th percentile). The significance of differences in the changes from baseline to 24 weeks between the continue group and switch group was examined using an unpaired *t*-test or the Mann–Whitney U test. * *p* < 0.05, ** *p* <0.01, *** *p* < 0.001 vs. baseline via a paired *t*-test or Wilcoxon’s signed-rank test. γ-GTP, γ-glutamyl transpeptidase; ALT, alanine aminotransferase; AST, aspartate aminotransferase; BP, blood pressure; DTSQ, diabetes treatment satisfaction questionnaire; eGFR, estimated glomerular filtration rate; FPG, fasting plasma glucose; HbA1c, glycated hemoglobin; HDL, high-density lipoprotein; TDI, total daily insulin; wks, weeks. ^a^ Data from 24 subjects (Continue, *n* = 11; Switch to semaglutide, *n* = 13) treated with insulin. ^b^ Data from 57 subjects (Continue, *n* = 26; Switch to semaglutide, *n* = 31).

**Table 3 pharmaceutics-15-02163-t003:** Relationship between percent changes in the fatty liver index and baseline clinical parameters.

	Continue (*n* = 27)		SWITCH (*n* = 31)	
	Correlation Coefficient	*p*-Value	Correlation Coefficient	*p*-Value
Age	0.011	0.958	−0.429	0.016
HbA1c	−0.203	0.309	0.082	0.659
AST	−0.093	0.645	0.162	0.383
ALT	−0.170	0.369	0.314	0.085
eGFR	−0.015	0.942	0.099	0.595
Total cholesterol	0.195	0.331	−0.191	0.304
Fatty liver index	−0.190	0.342	0.4912	0.005

*p*-values were obtained via Spearman’s rank correlation analysis. ALT, alanine aminotransferase; AST, aspartate aminotransferase; eGFR, estimated glomerular filtration rate; HbA1c, glycated hemoglobin.

## Data Availability

The data that support the findings of this study are available from the corresponding author upon reasonable request.

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
