# Peer review of "Effect of Switching to Once-Weekly Semaglutide on Non-Alcoholic Fatty Liver Disease: The SWITCH-SEMA 1 Subanalysis"

_pharmaceutics, 2023, doi:10.3390/pharmaceutics15082163_

Round 1
Reviewer 1 Report
The current manuscript, submitted herein presents clinically promising treatment protocol for better management and control of NAFLD. Of the note, the limitations of this study were also presented so that the message is broad and informative for future researchers.
However, there are some points of concern, that would, if addressed, make the data and conclusion more informative and more convincing.
It is evident that change in FLI was significant when switching from Dulaglutide to semaglutide as compared to switching from Liraglutide in case of de, which was not significance with p value 0.55 as in Figure 3. It is even you can barely/hardly say there is a trend. The known trend when P value more than 0.05 up to 0.1. But it can be said, as the authors reported, that the decrease in FLI (improvement) is higher in switching from Dulaglutide than in case of switching from Liraglutide to Simaglutide.
1- Th authors should comment on this point, mentioned above (differential response) in discussion.
Among points of critical thinking for the comment on this differential response: The number of patients (n) in in figure 3 should be stated in the legend. This may clear, at least, partially, if higher n was in case of Dulaglutide than in Liraglutide. if n is close such as 15, 18.
In general, (n) should be stated in all legends (29 continuous, 33 switch to Semaglutide) as it is done in tables.
One suggested sub-analysis to figure out if this difference in response between switching from Liraglutide vs from Dulaglutide: If baseline and 24 weeks values of ¥-GTP, TG, waist circumference (WC) all listed for the two sub-groups (switch from Liraglutide vs switch from Dulaglutide) and run t-test to see if the baseline read out of these parameters (elements in FLI index calculation) are significantly different or at least showing strong trend p < 0.1 difference at baseline then it is pre-existing difference due to different GLP-R1As.
If the p value of this test is high with almost small or no difference, then the differential response is due to possible higher samples (n) in Dulaglutide and/or other mechanistic difference between the two GLP-1Ras (Lira and Dula).
One more point of analysis for the same question: age for each patient can be aligned individually with fatty liver index (FLI) at base and 24weeks, in tables for correlation, separately for the subgroup of “switch from Liraglutide” and “switch from Dulaglutide” and see if the age correlates strongly and significantly with FLI in “Switch from Dulaglutide” sub-group but not in “switch from Liraglutide” sub-group, then it can be explained based on age. Also, the wo sub-groups FLI can be listed in one table and tested for correlation with difference in FLI to see if the correlation will be positive or negative correlation with low or high p value.
- If the recommended analysis, above showed that the more reduction (improvement) in FLI in switch from Dulaglutide sub-group than in switching from Liraglutide sub-group is NOT or does Not seem to be a reflection to difference in age or difference in baseline elements of FLI, then this differential response in switching should be indicated /added clearly in the conclusion in an appropriate statement, and also recommended to be further investigated/confirmed in future studies for future directions at the end of discussion.
- One last and important approach to explain and properly cover this point in discussion is the published paper in Diabetes Obes Metabolism 2018 (2019 Jan; 21(1): 43–51.
- Published online 2018 Aug 23. doi: 10.1111/dom.13479).
Pasting from the conclusion of that paper Overgaard R.V. et al. 2018:
“” In summary, pharmacokinetic and pharmacodynamic differences exist between liraglutide, dulaglutide and exenatide ER, at the approved dose levels, and it is relevant to consider these when switching to semaglutide. Exposure‐response modelling can facilitate simulation of HbA1c and weight loss outcomes following a switch from other GLP‐1RAs to semaglutide treatment. Significant and clinically relevant improvements in HbA1c and body weight are expected to occur following a switch from any of the other GLP‐1RAs to semaglutide””
Citing this paper, taking its findings and summary to further explain this relevant point in the current submitted paper is extremely important to make this paper more informative and comprehensive.
2- SE should be expressed on the bars, instead of SD as it will make the statistically significant difference, specially in Figure 2, more visually matching with graphics of the figure for the readers.
Reviewer 2 Report
Journal: Pharmaceutics
Manuscript ID: pharmaceutics-2551545
Title: “Effect of switching to once-weekly semaglutide on non-alcoholic fatty liver disease: The SWITCH-SEMA 1 subanalysis”
Authors: Hiroshi Nomoto et al.
The authors of this article investigated the impact of once-weekly semaglutide on non-alcoholic fatty liver disease (NAFLD) by analyzing data from a multicenter, prospective, randomized, parallel-group trial. The trial involved comparing the switch from liraglutide or dulaglutide to once-weekly semaglutide in Japanese adults with type 2 diabetes over 24 weeks. The study findings revealed that the semaglutide group showed significant improvement in the Fatty Liver Index (FLI) score compared to the group receiving conventional daily GLP-1RAs. Specifically, the improvement in FLI was more pronounced when transitioning from dulaglutide to semaglutide, and it exhibited a strong correlation with older age and lower baseline FLI levels. Additional indices were also estimated. This study is of particular interest as it focuses on a medically significant condition that requires further research and treatment options. However, there are several major points that need to be further considered. Please see my comments below:
Comments:
1. In the introduction and/or discussion, the authors should justify their focus on the FLI index in the abstract and the main manuscript. It is essential to explain the reasons and rationale behind this choice, especially considering that other widely used and well-established indices were also assessed.
2. Please provide more detailed information regarding the inclusion and exclusion criteria of the study participants and group them. This should include a clear explanation of how other potential causes of fatty liver disease were excluded and identified, as well as any drugs that might affect intrahepatic triglyceride content, steatosis, or NAFLD in general. Additionally, please briefly clarify the reasons for excluding habitual drinkers.
3. In the materials and methods section, please define the study's outcomes, both primary and secondary if applicable.
4. Please rephrase the statistical analysis to improve the clarity of the content. Specifically, describe how the authors presented and statistically compared the continuous variables within groups based on their normal distribution. Additionally, mention the method used to assess the normal distribution.
5. Moreover, it would be beneficial to explain how the authors handled missing values and outliers in their analysis.
6. Please update Table 1 to include the presence of comorbidities, diabetes-related complications, smoking, and physical exercise.
7. The authors should also consider adjusting their findings for the delta change in BMI and HbA1 levels. This adjustment could provide further insights into the relationship between the examined variables and the indices.
8. Please briefly provide additional information regarding the dosage schedule (dose escalation/titration) and the final dose of semaglutide. It is noted that all participants received dulaglutide at a dose of 0.75 mg/week, which, according to the authors, is the only dose approved in Japan. Additionally, 9 out of 11 participants in the "continue" group and 9 out of 14 participants in the "switch" group received liraglutide at a daily dose of 1.2 mg or lower. Further elaboration on this point would be appreciated. Furthermore, could you please provide clarification regarding the daily doses of liraglutide, specifically the 0.9 mg and 1.5 mg doses administered to the participants as detailed in Table 1? Furthermore, according to the authors, weekly semaglutide demonstrated potent effects on glycemic control and body weight compared with liraglutide and dulaglutide. However, it is important to ascertain whether the effects of semaglutide on NAFLD are primarily driven by its impact on body weight and glycemic changes rather than being specific to this drug. Did the authors conduct an examination to explore this aspect? Additionally, it would be valuable to consider whether the administered doses of the daily GLP1s accurately reflect the maximum effect of these drugs on NAFLD. Were the doses among the participants in the three groups pharmaceutically equivalent in terms of their effectiveness? The interpretation of the findings should take these points into account and further discuss this issue in the discussion section.
9. In the discussion, the authors noted that "Considering that the magnitude of the improvements in HbA1c and BMI was similar under both switching strategies (liraglutide to semaglutide or dulaglutide to semaglutide) in this subanalysis (Table S2), potential differences in the underlying molecular mechanism of liraglutide and dulaglutide, especially regarding intrahepatic metabolism, could exist." It is important to consider whether this finding may have been influenced by the fact that the necessary number of participants (power calculation) was not achieved in this analysis. The limitation of not achieving the necessary number of participants in the analyses should also be taken into account when interpreting the results and further highlighted in the limitations paragraph.
10. As mentioned by the authors, "physical assessment markers and treatment satisfaction were evaluated at baseline and the end of the study (24 weeks)." To provide a comprehensive view, please update Table 2 to include this information. Additionally, the authors mentioned that "Diet and exercise regimens were continued appropriately." It would be helpful to know if the authors assessed the diet followed by the study participants and whether there were any potential differences between the two groups.
11. Please enrich the conclusion paragraph to briefly reflect and summarize the main findings and key take-home messages presented in this paper.
Minor:
1. Please ensure that all abbreviations used in the manuscript, including tables and figures (e.g., wks, etc.), are defined clearly to aid the readers' understanding.
Reviewer 3 Report
I have studied carefully the manuscript entitled "Effect of switching to once-weekly semaglutide on non-alcoholic fatty liver disease: The SWITCH-SEMA 1 subanalysis" by Nomoto H. et al.
The manuscript concerns a topic with growing interest, namely this of the potential beneficial effect of GLP-1 analogues on non-alcoholic fatty liver disease. Thus, it is believed that such a manuscript could reach a wide readership of specialists (endocrinologists, gastroenterologists) as well as general practitioners.
However, before considering publication, the authors are wellcome to discuss the following issues:
Major issue
1. As stated in Table 2, the two groups ("continue" and "swich to semaglutide") are not comparable in terms of BMI (p<0.001) and HbA1c (p<0.001). This immediately raises the question wether the reported improvement after switch to semaglutide is a per se (direct) effect or could be explained by the disadvantageous profile of the "switch to semaglutide" group. To further enlighten this point, the authors are wellcome to add BMI and HbA1c in their multiple linear regression model (Table S3).
2. The authors are wellcome to clearly report the post hoc power of the study.
Minor editing of English language required.
Round 2
Reviewer 2 Report
Journal: Pharmaceutics
Manuscript ID: pharmaceutics-2551545 (Revised Version)
Title: “Effect of switching to once-weekly semaglutide on non-alcoholic fatty liver disease: The SWITCH-SEMA 1 subanalysis”
Authors: Hiroshi Nomoto et al.
The authors have satisfactorily responded to my comments and suggestions. They have also implemented the necessary changes by significantly improving the paper's content and quality. There are no further considerations.
Reviewer 3 Report
I have studied the revised manuscript entitled "Eddect of switching to once-weekly semaglutide on non-alcoholic fatty liver disease: The SWITCH-SEMA 1 subanalysis" by Nomoto et al.
The authors have put substantial effort to ameliorate the manuscript. Indeed, many issues raised by the reviewers have been successfully resolved. However, the "conclusion" paragraph of the "Discussion" section is still elusive and request the strongest evidence possible to be further supported. Thus, before considering publication, the authors are wellcome to disuss the following points:
Major points
1) The authors have modified their inclusion/exclusion criteria; thus, the sample size has been further reduced, and so has the study's power. The authors are wellcome to numerically assess the post-hoc power of the study.
2) Analyzing the between-subjects variability, the authors have demonstrated that switching from liraglutide or dulaglutide to semaglutide, especially a switch from dulaglutide to semaglutide, might have preferable effects on liver steatosis and metabolic abnormalities such as glycemic control and body weight management. However, a major limitation is the potential underestimation of within-subjects variability, especially as far as BMI, HbA1c, and blood pressure are concerned. The repeated-measures ANCOVA (General Linear Model) efficiently analyzes between-subjects and within-subjects variability of the same variable measured more than once in each patient. The authors are wellcome to explicitly describe within- and between- subject variability concerning the dependent variables of interest (such as FLI index), using the repeated-measures ANCOVA on all included patients (n=58). This model can efficiently explain if the difference between two consequent measures in the same 58 subjects (at the beginning and at the end of the study; within subjects variability) might be attributed to the between-subject variability due to either the effect of swich to semaglutide (considered as binary parameter of type yes/no) or to any potential confounder (such as initial GLP-1, age, BMI, HbA1c, and blood pressure).
